# Platyphyllenone Induces Autophagy and Apoptosis by Modulating the AKT and JNK Mitogen-Activated Protein Kinase Pathways in Oral Cancer Cells

**DOI:** 10.3390/ijms22084211

**Published:** 2021-04-19

**Authors:** Yen-Tze Liu, Hsin-Yu Ho, Chia-Chieh Lin, Yi-Ching Chuang, Yu-Sheng Lo, Ming-Ju Hsieh, Mu-Kuan Chen

**Affiliations:** 1Oral Cancer Research Center, Changhua Christian Hospital, Changhua 500, Taiwan; 144084@cch.org.tw (Y.-T.L.); 183581@cch.org.tw (H.-Y.H.); 181327@cch.org.tw (C.-C.L.); 177267@cch.org.tw (Y.-C.C.); 165304@cch.org.tw (Y.-S.L.); 2Institute of Medicine, Chung Shan Medical University, Taichung 402, Taiwan; 3Department of Family Medicine, Changhua Christian Hospital, Changhua 500, Taiwan; 4Department of Holistic Wellness, Mingdao University, Changhua 52345, Taiwan; 5Post Baccalaureate Medicine, National Chung Hsing University, Taichung 402, Taiwan; 6Graduate Institute of Biomedical Sciences, China Medical University, Taichung 404, Taiwan; 7Department of Otorhinolaryngology, Head and Neck Surgery, Changhua Christian Hospital, Changhua 500, Taiwan

**Keywords:** platyphyllenone, autophagy, apoptosis, oral

## Abstract

Platyphyllenone is a type of diarylheptanoid that exhibits anti-inflammatory and chemoprotective effects. However, its effect on oral cancer remains unclear. In this study, we investigated whether platyphyllenone can promote apoptosis and autophagy in SCC-9 and SCC-47 cells. We found that it dose-dependently promoted the cleavage of PARP; caspase-3, -8, and -9 protein expression; and also led to cell cycle arrest at the G2/M phase. Platyphyllenone up-regulated LC3-II and p62 protein expression in both SCC-9 and SCC-47 cell lines, implying that it can induce autophagy. Furthermore, the results demonstrated that platyphyllenone significantly decreased p-AKT and increased p-JNK1/2 mitogen-activated protein kinase (MAPK) signaling pathway in a dose-dependent manner. The specific inhibitors of p-JNK1/2 also reduced platyphyllenone-induced cleavage of PARP, caspase-3, and caspase -8, LC3-II and p62 protein expression. These findings are the first to demonstrate that platyphyllenone can induce both autophagy and apoptosis in oral cancers, and it is expected to provide a therapeutic option as a chemopreventive agent against oral cancer proliferation.

## 1. Introduction

The causes of oral cancer are varied and complex. Common risk factors include tobacco and alcohol use, human papillomavirus (HPV) infection**,** and chewing betel nuts. Oral bacteria such as *Streptococcus anginosus* are also related to oral cancer and can promote the progression of oral cancer through chronic inflammation, suppression of cell apoptosis, and inhibition of the immune system [1]. Huang et al. indicated that over 80% of oral cancer patients in Taiwan are younger than 65 years old, and most of them are men [2]. The five-year survival rate of oral cancer decreases as the stage increases [2]. Studies have pointed out that early intervention and early detection of oral cancer can prolong life expectancy [2,3]. Chemotherapy and surgery are currently the main treatments for oral cancer. However, new medications and therapeutic strategies are warranted.

Platyphyllenone is a diarylheptanoid isolated from *Alnus nepalensis* leaves, and a study using leaves grown in India reported that this compound has antifilarial efficacy [4]. The leaf, roots, and bark of *Alnus nepalensis* are used in treating many illnesses such as dysentery, stomach ache, and diarrhea [4]. Diarylheptanoids exert anti-pancreatic cancer activity via decreased shh-Gli-FoxM1 pathway, induced cell cycle arrest, and suppressed cell proliferation [5]. In MCF-7 breast cancer cells, platyphyllenone was reported to induce apoptosis and cell cycle arrest by activating the JNK and p38 signaling pathways through reactive oxygen species (ROS) generation [6]. Other evidence shows that diarylheptanoids derived from different varieties also have chemoprotective, antioxidant, and anti-inflammatory properties in human lymphocyte DNA and a rat paw edema model [7,8].

Apoptosis was previously referred to as programmed cell death (PCD). It plays a vital role in embryonic development and maintenance of cells’ normal functions. In contrast to necrosis, which results in severe damage to cell organelles and a loss of cell membrane integrity, apoptosis causes minimal damage to surrounding cells and tissues [9] and does not cause inflammation [10]. The key features of apoptosis include the following: (1) The cell shrinks and becomes denser; (2) The concentrated chromatin is broken into many fragments (chromatin bodies); (3) There is little or no swelling of other organelles and mitochondria; (4) The DNA breaks down into multiples segments; (5) The cell breaks down into multiple small vesicles (apoptotic bodies); (6) The small vesicles are swallowed by macrophages to complete the process of apoptosis [10,11]. When cells are subjected to environmental damage or viral infections, which may harm the genetic material, the process of apoptosis is initiated to eliminate these dangerous cells that may cause cancer, thus saving the entire organism. In addition to apoptosis, autophagy is another mechanism that can induce DNA damage and cause cancerous cell death [12]. When a cell is subjected to stress, various proteins and lipids in the cell membrane form a phagophore and wrap the damaged organelle or protein. The phagophore will eventually be closed to the autophagosome. This autophagosome can be likened to a garbage truck in the cell, carrying waste to a disposal site, in this case the lysosome [13]. The outer membrane of the autophagosome is combined with the lysosome membrane, and the fusion forms an autolysosome. By means of hydrolytic enzymes in the lysate, the old organelles or proteins are broken down into small molecular substances. Other small molecular substances such as amino acids can be recycled into materials to complete autophagy [14,15,16]. In the process of autophagy, microtubule-associated protein 1A/1B-light chain 3 (LC3) is combined with phosphatidylethanolamine (PE) and is lipidated to form LC3-II, which then appears on the autophagosome membrane [17]. LC3-II is also the most common marker for detecting autophagy induction. In this study, we determined the anticancer effect of platyphyllenone on human oral cancer cells and explored its potential anticancer mechanisms. Our results demonstrated that platyphyllenone can induce autophagy and apoptosis by modulating the AKT and JNK pathways in oral cancer cells.

## 2. Results

### 2.1. Platyphyllenone Induces Cytotoxicity and Inhibits Colony Formation in Oral Cancer Cells

The chemical structure of platyphyllenone is shown in Figure 1A. To assess the cytotoxic effects of platyphyllenone on oral cancer cells, we treated SCC-9 and SCC-47 cells with different concentrations of platyphyllenone (0–40 μM) for 24, 48, and 72 h. The cell viability of SCC-9 and SCC-47 cells is depicted in Figure 1B. The results shown significantly decreased in a time- and dose-dependent manner in both cells. Then, we performed a colony formation assay of SCC-9 and SCC-47 cell lines, which had been treated with platyphyllenone. The results showed that the colony forming ability of two cell lines was suppressed compared with the control group (Figure 1C). Quantitative analysis of colony formation can even show the difference (Figure 1D).

### 2.2. Effect of Platyphyllenone on Cell Cycle Progression in Oral Cancer Cells

To preliminarily reveal the mechanism of growth suppression in the platyphyllenone-treated group, we treated SCC-9 and SCC-47 cells with different concentrations of platyphyllenone (0–40 μM) for 24 h and performed a cell cycle test through flow cytometry (Figure 2A). The results showed that platyphyllenone led to S phase accumulation both in SCC-9 and SCC-47 cells (Figure 2B). Moreover, we further evaluated the molecular basis of the cell-cycle-arrest-related protein through Western blotting. As shown in Figure 3C,D, for SCC-9 and SCC-47 cells after treatment with platyphyllenone (0–40 μM) for 24 h, the expression of cyclin A, cyclin B, CDK4, and CDK6 decreased significantly in high concentrations of 40 μM.

### 2.3. Platyphyllenone Induces Cell Apoptosis in Oral Cancer Cells

To clarify whether the inhibitory effect of platyphyllenone on the viability of oral cancer cells is related to the induction of apoptosis, we treated SCC-9 and SCC-47 cells with different concentrations of platyphyllenone (0–40 μM) for 24 h. Figure 3A shows Annexin-V and PI double-staining in the two cell lines. The quantitative graph shows that platyphyllenone significantly promotes apoptosis in SCC-9 and SCC-47 cell lines (Figure 3B). We next tested the DNA-binding dye DAPI to observe the changes in the nucleus with a fluorescence microscope in the two cell lines (Figure 3C). As shown in Figure 3C,D (quantitative fluorescence count), the nucleus has smaller cracked bubbles, which are sparsely dispersed in the SCC-9 and SCC-47 cells in high concentration of platyphyllenone. The results demonstrated that platyphyllenone-induced cytotoxicity can promote the phenomenon of apoptosis.

### 2.4. Platyphyllenone Activates Apoptosis via the Mitochondrial and Death Receptor Pathways and Regulators of Apoptosis-Related Proteins in Oral Cancer Cells

Many members of the Bcl-2 family bind to the mitochondrial membrane and can create many holes on the surface of the mitochondria. The formation of these holes causes the mitochondrial interstitium to release cytochrome c and causes the cell mitochondrial membrane potential (MMP) to change. Therefore, we first arranged SCC-9 and SCC-47 cells undergoing mitochondrial membrane potential analysis in flow cytometry analysis and quantified the percentage of cells going through depolarization (Figure 4A,B). The early depolarization in platyphyllenone 40 μM was seen in 81.45% and 31.68% of SCC-9 and SCC-47 cells, respectively (Figure 4B). Platyphyllenone can activate apoptosis through increasing cleaved poly (ADP-ribose) polymerase (PARP), cleaved caspase-3, cleaved caspase-8, cleaved caspase-9 in SCC-9 and SCC-47 cell lines, as confirmed by Western blot assay (Figure 4C,D). In Figure 4E,F, the results revealed that the extrinsic pathways of death receptor FAS, DR5, DCR3, DR2 proteins were significantly increased in platyphyllenone-treated SCC-9 and SCC-47 cells in a dose-dependent manner. Finally, the Bcl-2 family-related proapoptotic proteins Bax, Bak, BimL, BimS were increased in the platyphyllenone-treated group compared to the control group, and antiapoptotic protein Bcl-2 showed an opposite decreased expression (Figure 4G,H). These results establish that platyphyllenone promotes the apoptosis mechanism of SCC-9 and SCC-47 cells by activating different apoptotic pathways.

### 2.5. Platyphyllenone Induces Autophagy in SCC-9 and SCC-47 Oral Cancer Cells

Autophagy is an important protein degradation pathway, and it may also involve various diseases including cancer, neurodegenerative diseases, pathogen invasion, and muscle and liver diseases [18]. However, the effects of platyphyllenone on autophagy in human oral cancer has not been evaluated. We therefore sought to investigate whether SCC-9 and SCC-47 oral cancer cells treated with platyphyllenone (0–40 μM) for 24 h induced autophagy. We performed autophagy fluorescence analysis to observe formation of autophagosomes and Western blot assay to confirm the involvement of cytoplasmic microtubule-associated protein 1A/1B-light chain 3 and p62/SQSTM1 proteins in oral cancer cells. As shown in Figure 5A, the detection of autophagy in SCC-9 and SCC-47 increased with the platyphyllenone 40 μM treatment compared with the control group. The cytoplasmic microtubule-associated protein 1A/1B-light chain 3 (LC3) formations and sequestosome 1 (p62/SQSTM1) protein expression also increased after platyphyllenone treatment, which suggests that the formation of autophagosomes and promote autophagy effectively (Figure 5B–E).

### 2.6. The Induction of Apoptosis and Autophagy by Platyphyllenone Is Dependent on the Regulation of AKT and JNK Signaling Pathways in SCC-9 and SCC-47 Oral Cancer Cells

Mitogen-activated protein kinases (MAPKs) are important regulatory mechanisms in eukaryotic cells and are highly involved in the biological translation of cell autophagy and apoptosis [19,20]. Thus, we assessed the protein expression of MAPKs including p-JNK1/2, p-AKT, p-ERK1/2, and p-p38 in SCC-9 and SCC-47 cell lines treated with platyphyllenone through Western blot assay. The results revealed that p-AKT was decreased and p-JNK was increased both in SCC-9 and SCC-47 cells (Figure 6A,B). Accordingly, we pretreated both cell lines with specific inhibitors JNK and AKT (SP600125 and LY294002, respectively, 20 μM, each) for 1 h, then analyzed the combinatorial treatment with and without platyphyllenone for 24 h. The result indicates that a cotreatment of platyphyllenone and LY294002 increases the cleaved PARP and caspase-8 expression (Figure 6C,D), whereas the combination with SP600125 decreased expression (Figure 6E,F). As shown in Figure 6G–I, combination of LY294002 and SP600125 compared with platyphyllenone treatment alone significantly promoted and inhibited LC3-II expression. These findings suggest that the involvement of AKT and JNK1/2 activation may participate in regulation of platyphyllenone-induced apoptosis and autophagy.

## 3. Discussion

The occurrence of oral cancer has always been closely related to the habits of chewing betel nut, smoking, and drinking [21,22]. With the increase in the number of people chewing betel nut and smoking in Taiwan, the number of patients with oral cancer is also increasing [22]. Although there are many treatments for oral cancer, actively developing new therapeutic agents has always been an important issue. In this study, we demonstrated the effects of different platyphyllenone concentrations on the apoptosis and autophagy of two highly invasive oral cancer cell lines (SCC-9 and SCC-47). The SCC-47 cell lines also contain 18 copies of integrated HPV-16, which is correlated with the incidence of oropharyngeal squamous cell carcinoma (OPSCC) [23,24,25]. Notably, the incidence of head and neck cancers related to alcohol, cigarettes, and tobacco has decreased in developed countries, while the number of HPV-positive head and neck carcinoma cases has been increasing [26]. In the past 15 years, the population with HPV-positive oropharyngeal cancer has been growing rapidly in Taiwan by approximately 30% [27]. HPVs are also a high-risk factor for gynecological cancer [28]. Our results demonstrated that platyphyllenone can cause toxic damage and can induce apoptosis and autophagy in both HPV-positive and -negative cases. Future studies should evaluate the application of this agent in the treatment of oral cancer or other HPV-positive cancers.

In our study, platyphyllenone has cytotoxic effects on SCC-9 and SCC-47 cells with different doses (0–40 μM) according to the MTT assay (Figure 1B). The clonogenic assay were first performed to examine oral cancer cell growth into colonies on a single cell (Figure 1C,D). Platyphyllenone has the ability to inhibit cell proliferation at concentrations of 0–40 μM, consistent with previous studies on pancreatic cancer [5]. Next, we found that platyphyllenone induced S phase cell cycle arrest both in SCC-9 and SCC-47 cell lines (Figure 2A,B). Other studies indicated that diarylheptanoids revealed induction of S-phase cell cycle arrest and induced apoptosis by activation of caspases-3 and -9 in human neuroblastoma cells [29]. Past studies have shown that cyclin-dependent kinase 4 and 6 (CDK4 and CDK6) are generally overexpressed in various tumor cells and related to the increase of cyclin A in head and neck carcinomas [30]. Our study also found that platyphyllenone can effectively reduce cyclin A and B and CDK 4 and 6 (Figure 2C,D). Moreover, we demonstrated through flow cytometric and microscopic data that platyphyllenone exerts cytotoxicity by inducing apoptosis (Figure 3A–D). This evidence suggests that platyphyllenone induces oral cancer cell apoptosis and effectively regulates cell cycle apparatus.

Recent research on cancers often focuses on autophagy and apoptosis together [31,32,33]. As shown in Figure 4A, under fluorescence detection, the autophagosome formation of both SCC-9 and SCC-47 gradually increases with platyphyllenone in a dose-dependent manner. In oral cancers, many pure compounds and phytochemicals have been reported to induce cell autophagy effectively through increased LC3 and p62/SQSTM1 [34,35]. After 24 h of platyphyllenone treatment, our results showed that the expression of LC3I/II and p62/SQSTM1 was significantly increased in SCC-9 and SCC-47 cells (Figure 4B,C). This is also the first study to suggest that the platyphyllenone has the potential to promote autophagy in oral cancer.

MAPK pathways are a type of serine/threonine protein kinases widely found in prokaryotic and mammalian cells and are always involved in gene expression, cell division, differentiation, apoptosis, autophagy, and even cancer cell migration, invasion, and other carcinogenesis [36,37,38,39,40]. A previous study indicated that platyphyllenone and its derivative complex can increase reactive oxygen species (ROS) generation and activate JNK and p38 to further induce MCF-7 cell apoptosis [6].

Hence, after treatment using platyphyllenone, our data showed that it can increase the activation of JNK but also inhibit the expression of AKT in SCC-9 and SCC-47 cells (Figure 6A,B). To elucidate and verify these results, we applied SP600125 and LY294002 as JNK and AKT inhibitors, respectively, for further confirmation of manipulated apoptosis and autophagy. Our data revealed that co-treatment with platyphyllenone and SP600125 could strongly suppress cleaved PARP, cleaved caspase-3, and LC3I/II compared to the platyphyllenone-only group in both SCC-9 and SCC-47 cells (Figure 6E,F,I,J). Similar results were observed with platyphyllenone and LY294002 cotreatment (Figure 6C,D,G,H). Taken together, these findings imply that platyphyllenone induces apoptosis and autophagy in oral cancer cells via JNK and AKT signaling pathways.

## 4. Materials and Methods

### 4.1. Cell Culture

We used two human tongue cancer cell lines: SCC-47 and SCC-9. The SCC-47 cell line was isolated from the primary tumor of the lateral tongue of a man. It contains 18 copies of integrated HPV-16 and was obtained from Merck KGaA (Ann Arbor, MI, USA). SCC-9, the human tongue squamous carcinoma cell line, was obtained from ATCC (Manassas, VA, USA). The culture medium of SCC-47 is Dulbecco’s modified Eagle’s medium (DEME; Life Technologies, Grand Island, NY, USA) containing 1% nonessential amino acids (NEAA; Gibco, MA, USA) and 10% fetal bovine serum (Merck Millipore, MA, USA). SCC-9 cells were cultured in Dulbecco’s modified Eagle’s medium (DEME; Life Technologies) supplemented with Ham’s F12 Nutrient Mixture (Life Technologies), with 10% fetal bovine serum (Merck Millipore), hydrocortisone (400 ng/mL) (Sigma-Aldrich, Louis, MO, USA), and 1% NEAA. Both culture media contained 1.2 g/L sodium bicarbonate (Merck Sigma), 15 mM HEPES (Merck Sigma), and 1% penicillin/streptomycin (Gibco). All cells were maintained in a 5% CO_2_ atmosphere at 37 °C in a humidified incubator.

### 4.2. Platyphyllenone Treatment and Chemicals

Platyphyllenone was purchased from Chemfaces (≥98% purity). It was a phenol-type compound and was dissolved in dimethyl sulfoxide (DMSO). The platyphyllenone stock was prepared at a concentration of 100 mM and stored at −20 °C. The final experimental concentration treatments of DMSO content were consistently less than 0.1%. The 3-(4,5-dimethylthiazol-2-y1)-2,5- diphenyltetrazolium bromide (MTT) and diamidino-2-phenylindole (DAPI) were obtained from Sigma-Aldrich (St Louis, MO, USA). The apoptosis assay of cell cycle, Annexin V/PI Double Staining, and Mitochondrial Membrane Potential kits were purchased from Muse™, Millipore (Billerica, MA, USA). All antibodies were purchased from Cell Signaling Technology (Danvers, MA, USA). The specific inhibitors of SP600125 and LY294002 and were acquired from LC Laboratories (New Boston St, MA, USA). Bafilomycin A1 was acquired from MedChemExpress (Monmouth Junction, NJ, USA).

### 4.3. Cell Cytotoxicity Assay

SCC-9 and SCC-47 cells were distributed into 96-well plates with 1 × 10^4^ cells per well. After cultivation in the 37 °C and 5% CO_2_ incubator for 24 h until the cells were fully attached, the platyphyllenone treatment of 0, 20, 30, or 40 μM was then added in sequence and incubated for 24 h until the drug was absorbed. The culture medium was then removed, MTT agent (0.5 mg/mL) was added and mixed with the culture medium (1:9) for re-cultivation in an incubator at 37 °C and 5% CO_2_ for 4 h, and then the blue crystallized MTT was dissolved with methanol. The absorbance was determined at 595 nm by utilizing the spectrophotometer (BioTek Synergy HTX Multi-Mode Reader, Winooski, VT, USA). Each analysis was repeated in three separate experiments.

### 4.4. Colony Formation Assay

SCC-9 and SCC-47 cells were cultured in a 6-well plate with 5 × 10^2^ cells per well and then treated with 0, 20, 30, 40 μM platyphyllenone and slowly cultivated for 10 days. The culture medium was replaced once every 3 days during the experiment. The cells were fixed with methanol for 10 min, using Giesa (Merck Sigma) staining after methanol evaporated. Finally, the well count with colonies was photographed, and the survival was calculated.

### 4.5. Cell Cycle Analysis

SCC-9 and SCC-47 cells were cultured in a 6-well plate with an appropriate number of cells (3 × 10^5^/well) and then exposed to 0, 20, 30, 40 μM platyphyllenone for 24 h. After 1× PBS washing, the cells were fixed with 70% ethanol and frozen at −20 °C overnight prior to staining with Muse Cell Cycle Assay PI buffer (Millipore). The next day, the PI buffer was used for staining, and the cells were incubated for 30 min in the dark at room temperature. The buffer was mixed thoroughly, and analysis data from the Muse^®^ Cell Analyzer Assays (Millipore) was obtained. Each analysis was repeated in three separate experiments.

### 4.6. Annexin V/PI Double Staining Analysis

SCC-9 and SCC-47 cells were cultured in a 6-well plate with an appropriate number of cells (3 × 10^5^/well) and then exposed to 0, 20, 30, 40 μM platyphyllenone for 24 h. Briefly, the 1 × 105 cells were suspended in 100 μL PBS (contained 2% BSA), and 100 μL of Muse Annexin V & Dead Cell Reagent was added for a total of 200 μL solution in each tube. After incubation in the dark for 20 min at room temperature, Muse Annexin V & Dead Cell Analyzer analysis data was collected using Muse^®^ Cell Analyzer Assays (Millipore). Each analysis was repeated in three separate experiments.

### 4.7. Mitochondrial Membrane Potential Analysis

The analysis was conducted by using Muse Mitopotential Assay Kit (Merck Millipore). The SCC-9 and SCC-47 cells were collected at 1 × 10^5^ per tube and treated for 24 h with different concentrations of platyphyllenone (0, 20, 30, 40 μM). After PBS washing, Muse MitoPotential working solution was added to the sample, which was reacted at 37 °C for 20 min. Muse MitoPotential 7-AAD was added soon afterwards, reacting for 5 min at room temperature, and Muse MitoPotential Analyzer and analysis data were obtained using the Muse^®^ Cell Analyzer Assays (Merck Millipore). Each analysis was repeated in three separate experiments.

### 4.8. DAPI Staining Assay

SCC-9 and SCC-47 cells were cultured in a 6-well plate with an appropriate number of cells (3 × 10^5^/well) to indicate treatment. After the cells were fixed with 4% paraformaldehyde for 30 min, the DAPI reagent was dissolved (1:10,000) in triton-X100 mixed with 1× PBS solution and reacted for 15 min. Cells were then washed with PBS to remove the solution and photographed by using fluorescence microscopy (Lecia, Bensheim, Germany). Cell morphological changes were observed and quantified.

### 4.9. Western Blot Analysis

SCC-9 and SCC-47 cells were distributed with an appropriate number of cells (5 × 10^5^/dish) in 6 cm dishes. Cells were treated for 24 h with different concentrations of platyphyllenone (0, 20, 30, 40 μM) in a similar way. The process of collecting protein lysate and antibodies was as previously described [16]. The final signals were observed using ECL detection and photographed with a chemiluminescence fluorescence Image Quant LAS 4000 (GE Healthcare, Berlin, Germany) biomolecule imaging system. Image J software was used for protein quantification.

### 4.10. Autohphagy Fluorescence Analysis

SCC-9 and SCC-47 cells were distributed into a 96-well culture plate with an appropriate number of cells (1 × 10^4^/well). After treatment with platyphyllenone 0, 20, 30, 40 μM for 24 h, The Cell Meter™ Autophagy Kit (AAT Bioquest, Sunnyvale, CA, USA) was used to detect specific autophagosome markers with green fluorescence. The absorbance was determined at 488 nm by utilizing a spectrophotometer (BioTek Synergy HTX Multi-Mode Reader, Winooski, VT, USA). Each analysis was repeated in three separate experiments.

### 4.11. Statistical Analysis

All experiments were repeated at least three times, and quantitative data are expressed as the mean ± standard deviation (SD). GraphPad Prism Software Version 5.0 (GraphPad Software Inc., La Jolla, CA, USA) was used with Student’s *t* test for biostatistics graphing. A *p* value < 0.05 was considered statistically significant.

## 5. Conclusions

Collectively, this study is the first direct evidence that platyphyllenone has the ability to induce cell apoptosis and autophagy through reduction of AKT and enhancing JNK1/2 phosphorylation in head and neck squamous cell carcinoma. Platyphyllenone affects the oral cancer cell cycle and causes arrest in the S phase. Thus, platyphyllenone is expected to be the foundation for the development of analogs for treating oral cancer.

## Figures and Tables

**Figure 1 ijms-22-04211-f001:**
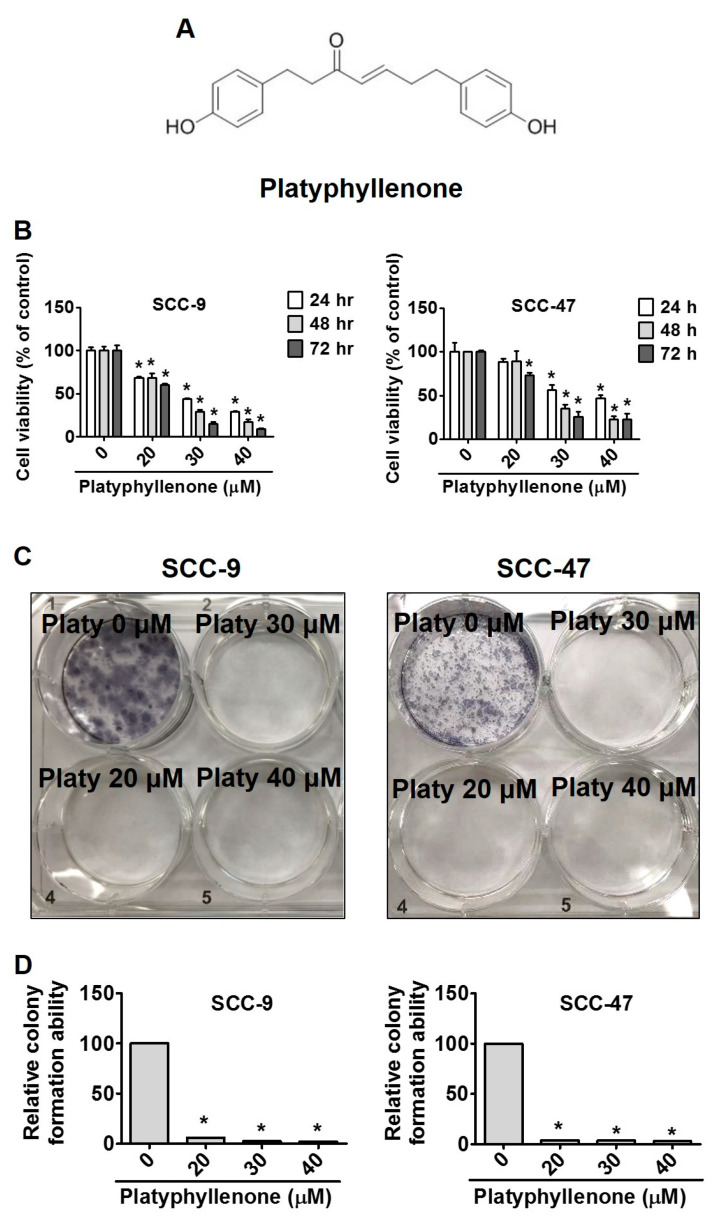
Platyphyllenone induces cytotoxicity and inhibits colony formation in oral cancer cell lines. (**A**) Chemical structure of platyphyllenone. (**B**) SCC-9 and SCC-47 cell lines were treated with different concentrations (0, 20, 30 and 40 μM) of platyphyllenone for 24 h, and the cell viability was determined by MTT assay. The values represent the means ± SD of at least three independent experiments. (**C**) After treatment with platyphyllenone, SCC-9 and SCC-47 were confirmed by colony formation assay. (**D**) Quantitative analysis of colony formation of SCC-9 and SCC-47 cells. Student’s *t* test was applied to determine statistical significance, and two-tailed *p* values are shown (* represents *p* < 0.05).

**Figure 2 ijms-22-04211-f002:**
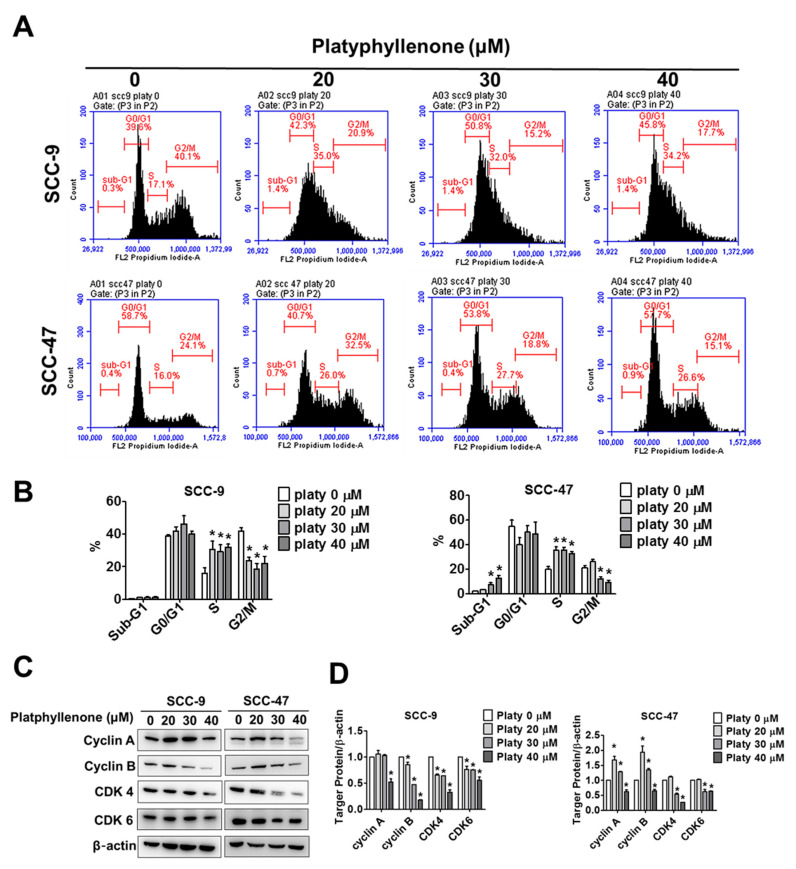
Effect of platyphyllenone on cell cycle progression in oral cancer cells. (**A**) The SCC-9 and SCC-47 cells were assessed by flow cytometry with different concentrations of platyphyllenone (0, 20, 30, and 40 μM) after 24 h. (**B**) The quantitative analysis of cell cycle at sub-G1, G0/G1, S, G2/M for SCC-9 and SCC-47 cell lines. (**C**) Western blot analysis of effect of platyphyllenone on cell cycle regulatory protein (including cyclin A, cyclin B, CDK 4, CDK 6) expression in SCC-9 and SCC-47 cells. (**D**) Quantitative results of (**C**) in SCC-9 and SCC-47 cells and each protein expression after being adjusted with β-actin. The values represent the means ± SD of at least three independent experiments. Student’s *t* test was applied to determine statistical significance, and two-tailed *p* values are shown (* represents *p* < 0.05).

**Figure 3 ijms-22-04211-f003:**
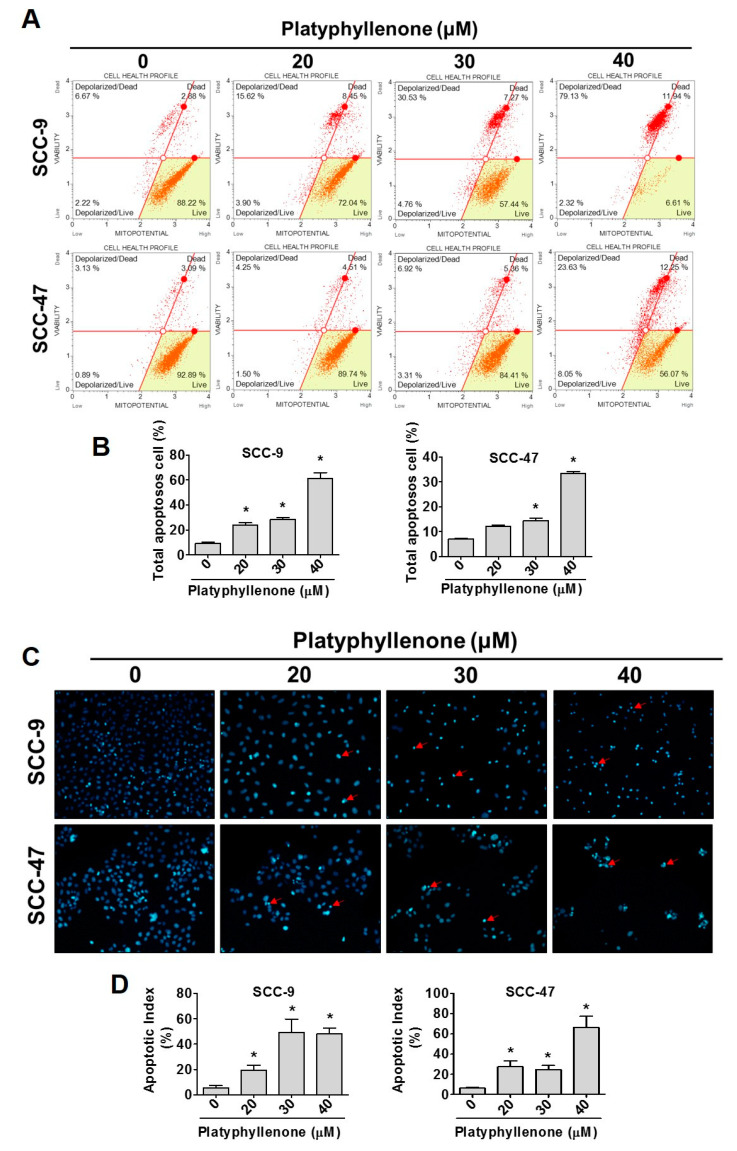
Platyphyllenone induces cell apoptosis in oral cancer cells. (**A**,**B**) SCC-9 and SCC-47 cell lines were treated with 0–40 μM of platyphyllenone and analyzed through flow cytometry after annexin-V/PI double staining, and the results of both cells were quantified. (**C**,**D**) SCC-9 and SCC-47 cells were observed by DAPI staining under a fluorescence microscope, and nuclei condensation and fragmentation (red arrow) were increased with platyphyllenone concentration. The values represent the means ± SD of at least three independent experiments. Sacle bar = 100 μm. Student’s *t* test was applied to determine statistical significance, and two-tailed *p* values are shown (* represents *p* < 0.05).

**Figure 4 ijms-22-04211-f004:**
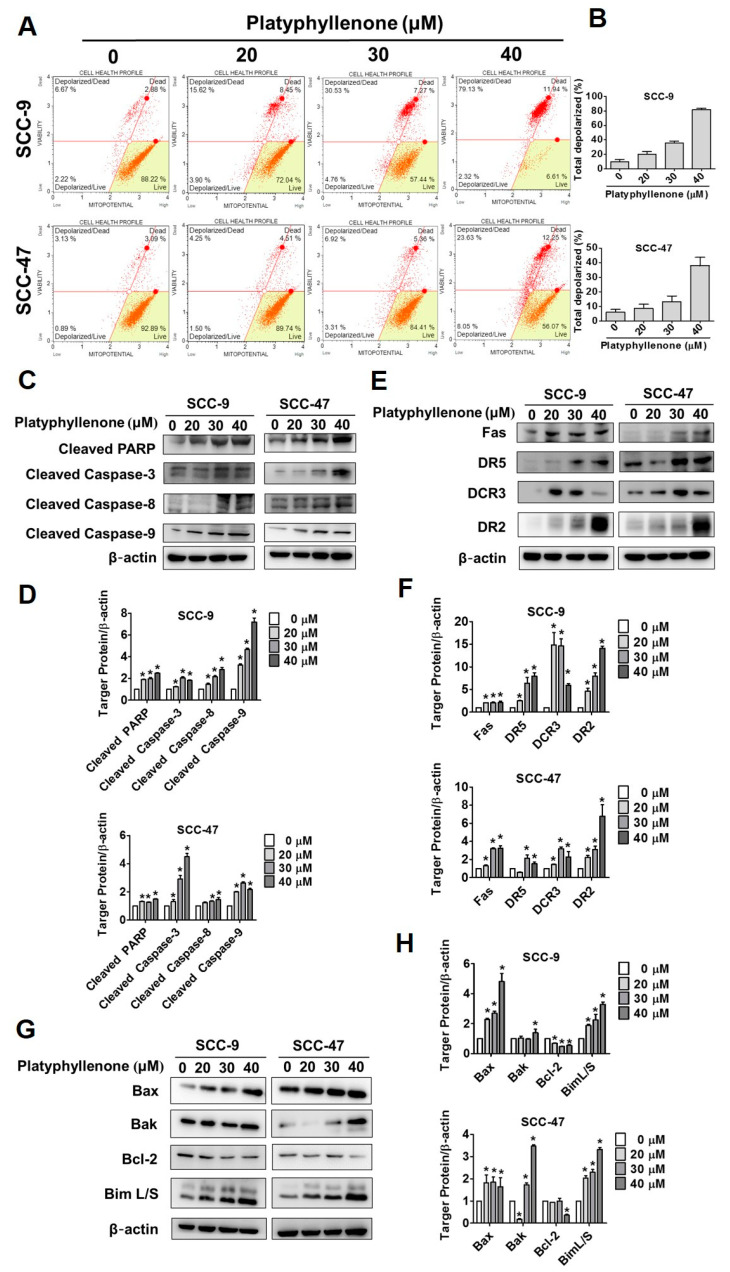
Platyphyllenone activates apoptosis via the mitochondrial and death receptor pathways and regulators of apoptosis-related proteins in oral cancer cells. (**A**,**B**) SCC-9 and SCC-47 cell lines were treated with different concentrations of platyphyllenone (0–40 μM) and using flow cytometry to analyze the effect on mitochondrial membrane potential, and the results of both cells were quantified. (**C**,**D**) After treatment with platyphyllenone on SCC-9 and SCC-47 cells, Western blotting was used to detect the protein expression of cleaved PARP, cleaved caspase-3, cleaved caspase-8, and cleaved caspase-9, and the results of both cells were quantified. (**E**,**F**) Western blotting detected the expression change in Fas, DR5, DCR3, and DR2, the results of SCC-9 and SCC-47 cells were quantified. (**G**,**H**) The above-described method was used to determine the expression changes in Bax, Bak, Bcl-2, and Bim L/S and quantify the results of SCC-9 and SCC-47 cells. All quantitative results are given for each protein expression after being adjusted with β-actin. The values represent the means ± SD of at least three independent experiments. Student’s *t* test was applied to determine statistical significance, and two-tailed *p* values are shown (* represents *p* < 0.05).

**Figure 5 ijms-22-04211-f005:**
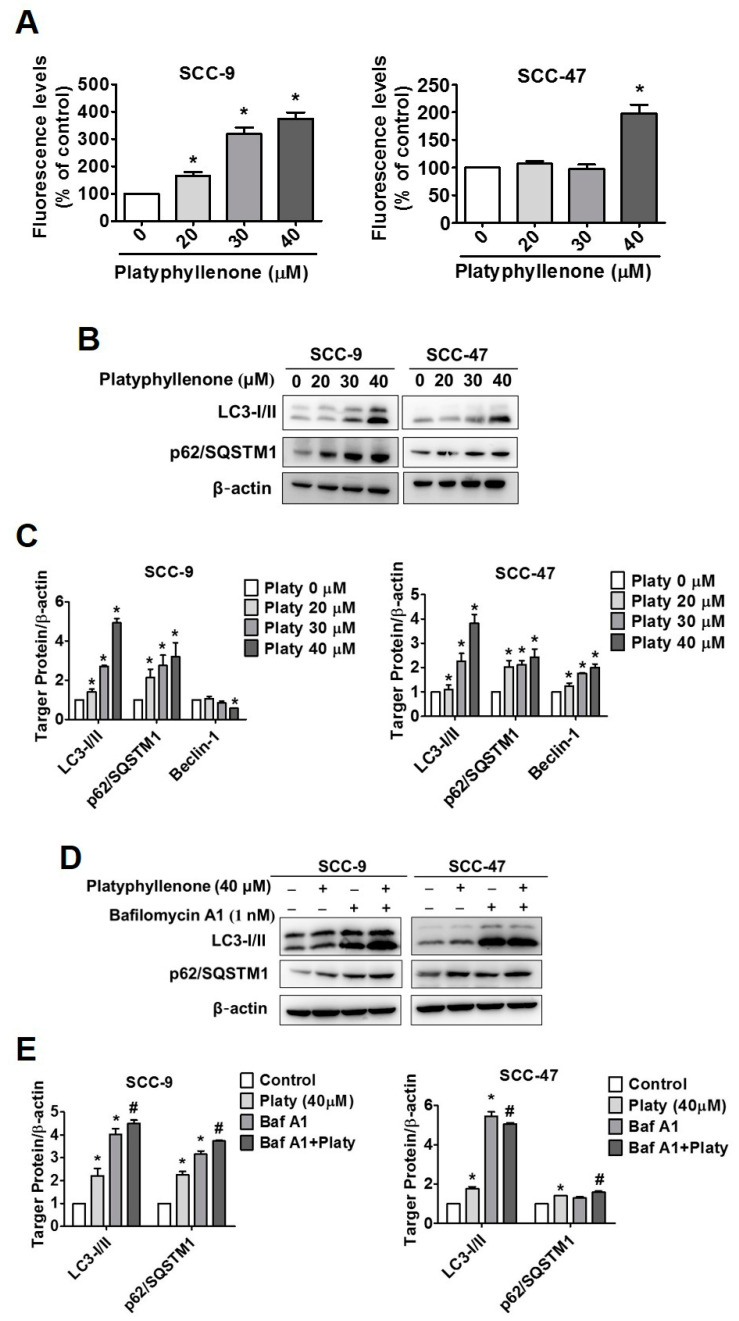
Platyphyllenone induces autophagy in SCC-9 and SCC-47 oral cancer cells. (**A**) SCC-9 and SCC-47 cells were treated with platyphyllenone after 24 h; the autophagosome marker with green fluorescence was detected using a spectrophotometer. (**B**,**C**) After treatment with platyphyllenone on SCC-9 and SCC-47 cells, Western blotting was used to detect the protein expression of LC3I/II and p62/SQSTM1, and the results of both cells were quantified. (**D**,**E**) After combined treatment with platyphyllenone and bafilomycin A1 on SCC-9 and SCC-47 cells, Western blotting was used to detect the protein expression of LC3I/II and p62/SQSTM1, and the results of both cells were quantified. “+”, treat; “−”, non-treat. All quantitative results are given for each protein expression after being adjusted with β-actin. The values represent the means ± SD of at least three independent experiments. Student’s *t* test was applied to determine statistical significance, and two-tailed *p* values are shown (* represents *p* < 0.05; # represents *p* < 0.05).

**Figure 6 ijms-22-04211-f006:**
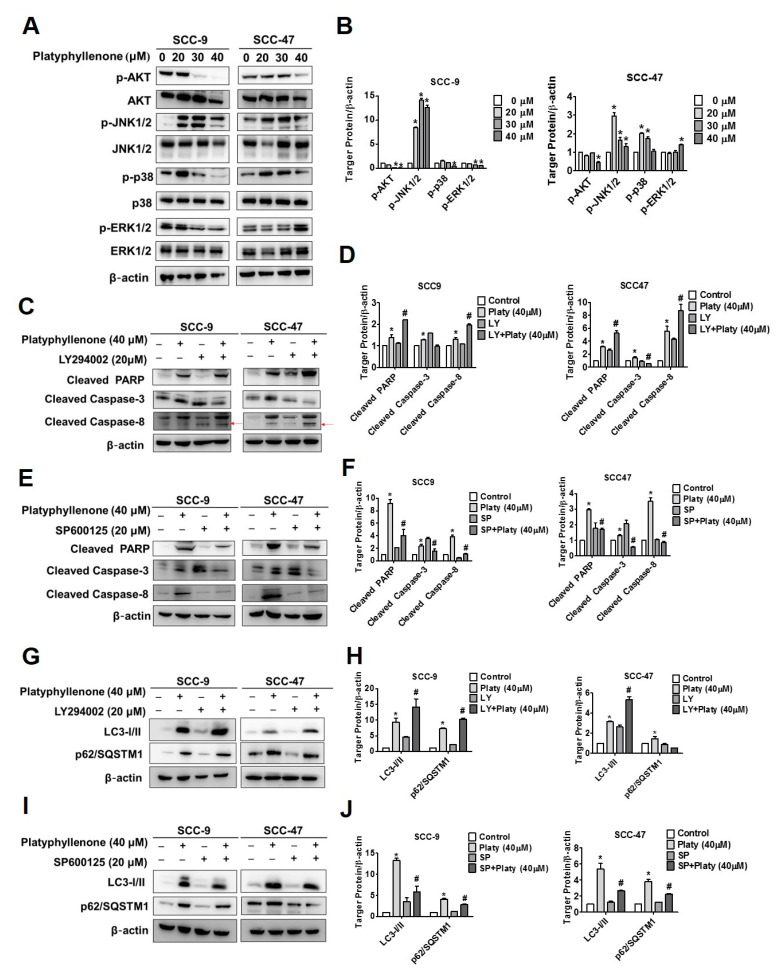
The induction of apoptosis and autophagy by platyphyllenone is dependent on the regulation of AKT and JNK signaling pathways in SCC-9 and SCC-47 oral cancer cells. “+”, treat; “−”, non-treat. (**A**,**B**) After treatment with platyphyllenone on SCC-9 and SCC-47 cells, Western blotting was used to detect the protein expression of p-AKT, p-JNK1/2, p-p38, p-ERK1/2, and the results of both cells were quantified. (**C**–**J**) SCC-9 and SCC-47 cells were pretreated with LY294002 or SP600125 for 1 h and then incubated in the presence or absence of 40 μM platyphyllenone for 24 h. Western blotting was then used to detect the protein expression of apoptosis (**C**–**F**) and autophagy (**G**–**J**) related proteins and quantify results. All quantitative results are given for each protein expression after being adjusted with β-actin. The values represent the means ± SD of at least three independent experiments. Student’s *t* test was applied to determine statistical significance, and two-tailed *p* values are shown (* represents *p* < 0.05; # represents *p* < 0.05).

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
