# Peer review of "Platyphyllenone Induces Autophagy and Apoptosis by Modulating the AKT and JNK Mitogen-Activated Protein Kinase Pathways in Oral Cancer Cells"

_ijms, 2021, doi:10.3390/ijms22084211_

Round 1

Reviewer 1 Report

Ho et al., have studied the “Platyphyllenone induce autophagy and apoptosis through modulation of AKT and JNK mitogen-activated protein kinase pathway in oral cancer cell”. The authors have evaluated the Platyphyllenone anti-cancer activity on oral cancer cells and found that the compound is competent enough to kill cancer via inducing apoptosis and autophagy.

Although the study and results are encouraging, lack of fluency in language dragging down.

First of all, the manuscript deserves extensive English corrections. I request authors to take help from the native speakers of English.

Line 17: “diarylheptanoids that may be provide with anti-inflammatory and chemoprotective effect.” ------ diarylheptanoids that exhibit anti-inflammatory and chemoprotective effects.

Line 24: “dose-dependent” ---- dose-dependent manner

Line 33: “Streptococcus anginosus” should be italics.

Line 48: “generatio” ---- generation

Line 66-79: including these two references would be appropriate. https://doi.org/10.1038/s41573-019-0036-1

https://doi.org/10.3390/cells9051321

Figure 2a: looks blurred; please increase the resolution and keep the common x and y axis, to enhance the appearance.

Figure 5: Increased LC3-II and p62 doesn’t necessarily increase the autophagy process, it happens even in the inhibition state. However, to confirm the same a functional control should be used Bafilomycin A1.  https://doi.org/10.3390/cells9051321

Figure 6: It is so heavy and not legible to read, please enlarge or divide.

Reviewer 2 Report

The authors present an investigation into the potential mechanisms of action of Platyphyllenone in oral carcinoma cells.  The resulting data is compelling evidence that this compound induces both apoptosis and autophagy in the cell lines examined.

I have no major concerns with this manuscript.  While the study itself is not particularly novel, the logical approach taken to answering the hypothesis does not leave much room for significant criticism.

Some points to consider:

  • The manuscript would benefit from close proofreading throughout.
  • The resolution of a number of the figures (especially the flow cytometry histograms) needs to be significantly enhanced throughout.  The reader should be able to read all the aspects of the histograms.
  • There should be a consistent approach taken throughout the manuscript to the size of graphs and western blots.  Some of the figures appear too spread out (e.g. fig 1 and 5), whereas others are too compacted (e.g. figure 6). Font size in figures is all over the place.
  • In the methods, please put the exponents of plated cell numbers into superscript.
  • Figure 3B- add significancy to graphs.
  • On figure 3C- add arrows to indicate examples of cells of interest.
  • In section 2.5, please mention the assay utilised to measure autohphagy.  In the current form, the reader needs to go to methods to see what this assay is called. Given that this is an image based assay, images of the cells +/- treatment would make a welcome addition to figure 5.

Round 2

Reviewer 1 Report

As requested, the authors have made substantial modifications to the manuscript. 

Hereby I endorse the manuscript for publication.